# ERF-RTMDet: An Improved Small Object Detection Method in Remote Sensing Images

Shuo Liu [1], Huanxin Zou [1,*], Yazhe Huang [2], Xu Cao [1], Shitian He [1], Meilin Li [1] and Yuqing Zhang [1]

1   College of Electronic Science and Technology, National University of Defense Technology, Changsha 410073, China; liushuo21@nudt.edu.cn (S.L.); cx2020@nudt.edu.cn (X.C.); heshitian19@nudt.edu.cn (S.H.); limeilin@nudt.edu.cn (M.L.); zhangyuqing22@nudt.edu.cn (Y.Z.)
2   Tianjin Advanced Technology Research Institute, Tianjin 300457, China; u201810086@alumni.hust.edu.cn
*   Correspondence: zouhuanxin@nudt.edu.cn; Tel.: +86-731-8700-3288

**Abstract:** A significant challenge in detecting objects in complex remote sensing (RS) datasets is from small objects. Existing detection methods achieve much lower accuracy on small objects than medium and large ones. These methods suffer from limited feature information, susceptibility to complex background interferences, and insufficient contextual information. To address these issues, a small object detection method with the enhanced receptive field, ERF-RTMDet, is proposed to achieve a more robust detection capability on small objects in RS images. Specifically, three modules are employed to enhance the receptive field of small objects' features. First, the Dilated Spatial Pyramid Pooling Fast Module is proposed to gather more contextual information on small objects and suppress the interference of background information. Second, the Content-Aware Reassembly of Features Module is employed for more efficient feature fusion instead of the nearest-neighbor upsampling operator. Finally, the Hybrid Dilated Attention Module is proposed to expand the receptive field of object features after the feature fusion network. Extensive experiments are conducted on the MAR20 and NWPU VHR-10 datasets. The experimental results show that our ERF-RTMDet attains higher detection precision on small objects while maintaining or slightly enhancing the detection precision on mid-scale and large-scale objects.

**Keywords:** small object detection; enhanced receptive field; remote sensing; dilated convolution; hybrid attention

## 1. Introduction

Due to its broad coverage and unique high-altitude view, object detection in RS (remote sensing) datasets has received widespread focus with practical applications in various fields, such as military surveillance, environmental analysis, and precision agriculture. Traditional object detection methods typically involve extracting image features by manual design. These features are then combined to classify objects. However, generating the candidate regions is time-consuming. Designing features manually is also unreliable, complex, and poorly generalized. These traditional methods have limited practical application. Therefore, the focus of research on object detection methods has gradually shifted to the application of deep learning methods.

Generally, object detection approaches that employ CNNs fall into two major types: two-stage and one-stage approaches, depending on different pipelines [1]. Specifically, the two-stage approaches, such as Faster R-CNN [2], conduct classification and regression following the generation of region proposals. The end-to-end pipeline of the one-stage approaches, such as YOLO [3], directly executes the classification and localization task on the output of the model's backbone. To sustain detection speed advantages while increasing accuracy, several attempts have been made with these approaches [4–8]. In addition to the above classification, mainstream object detection approaches are also categorized as

anchor-based or anchor-free methods, according to whether the priori bounding boxes are explicitly defined. Two-stage methods, like the proposals generated by RPN in Faster R-CNN [2], are generally anchor-based as they use candidate prior bounding boxes. One-stage methods used to be mainly anchor-based, but many anchor-free methods have been developed with satisfactory accuracy in recent years.

Anchor-based methods predict the location of objects of interest by matching the anchors with ground truth boxes. Representative methods, such as Mask R-CNN [9] and YOLOv2 [4], are employed. The R-CNN [10] implements a strategy of sliding windows to produce fixed anchors with predefined scales. However, utilizing the same group of anchors for instances of various sizes can reduce the effectiveness of object recognition. Therefore, it is necessary to set the corresponding hyperparameters of the anchors when applying anchor-based methods to different datasets for object detection. Achieving high detection accuracy is dependent on the proper hyperparameter settings. YOLOv2 [4] uses the K-means clustering method to obtain the size of the anchor box. In addition, the issue of imbalanced positive and negative samples impacts detection accuracy. This is because only a tiny proportion of the predefined anchors become positive samples. RetinaNet [11] provides a focal loss function that modifies positive/negative and hard/simple sample weights in the loss function using hyperparameters. YOLOv3 [5] sets nine distinct anchor boxes to predict multi-scale objects. Nevertheless, multi-scale objects with different orientations significantly increase the challenge of setting appropriate hyperparameters for anchors in high-altitude RS datasets.

Anchor-free methods predict objects by matching pixel points. The main advantages of anchor-free methods are their flexibility, generality, and low computational complexity. They avoid the massive computational resources typically required for anchors. Among them, the model's ability to capture small objects is enhanced by key-point-based regression methods. CornerNet [12], a pioneer in anchor-free methods, uses the upper-left and upper-right corners of the object bounding box as prediction key points. However, the key points used by CornerNet may fall outside the objects, causing the model to fail to capture internal information. This can lead to CornerNet missing instances when detecting small objects. Based on this, ExtremeNet [13] predicts four extreme points and one central point of the target. Another improvement of CornerNet, CenterNet [14], detects the central point of objects to predict the scales of bounding boxes. Unlike keypoint-based methods that rely solely on key points for detection, FCOS [15] uses a per-pixel approach. All anchors falling within the ground truth box are considered positive samples. The center-ness branch assigns a low score to the predicted boxes that deviate from the center of objects. YOLOX [16] adopts the center sampling strategy of FCOS. In addition, YOLOX improves detection accuracy by introducing a decoupling head, robust data augmentation, and a novel label assignment strategy. Based on YOLOX, RTMDet [17] achieves a further breakthrough in accuracy. It uses strategies such as deep convolution with a large kernel and dynamic soft label assignment. Therefore, we choose RTMDet as a robust baseline for the proposed method.

Current generic object detection approaches have developed various variants, such as increasing model width and depth and adding multi-scale fusion networks [18]. These methods are usually applied to natural images. The accuracy of these detection methods in RS images is not very satisfactory. This is because RS images have different characteristics from natural images. To illustrate this, some examples were selected from the MAR20 dataset. As seen in Figure 1a, RS images have a high density of small objects, which makes detection much more difficult. Second, objects in RS images vary significantly in scales and orientations, as illustrated in Figure 1b. This makes the detection task more challenging. Additionally, existing detection methods are prone to losing small objects' weak features in RS images. The complex scenes can interfere with detection, as Figure 1c shows. The overhead view of RS images reduces the available feature information for object detection. Small objects occupy a small resolution and are susceptible to interference from background information [19]. Consequently, the feature and contextual information

of small objects extracted by the model is limited [20]. Therefore, developing a method to enhance the effect of detecting small objects in RS datasets is urgently needed.

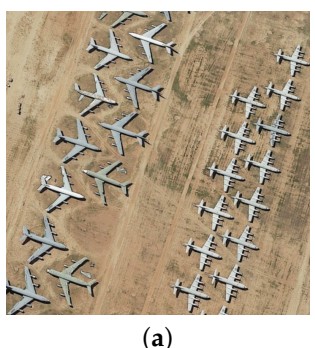 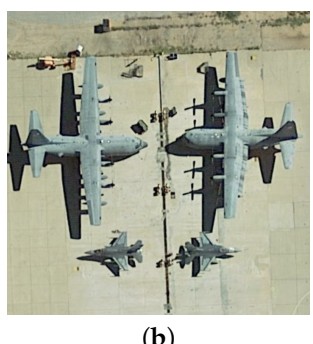 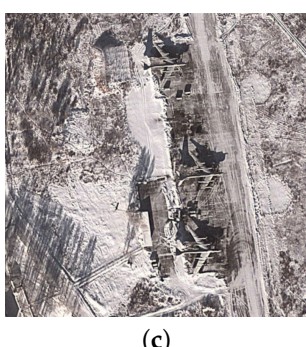

(**a**)                                   (**b**)                                  (**c**)

**Figure 1.** Several examples selected from the MAR20 dataset. (**a**) Densely arranged objects. (**b**) Objects with different scales and orientations. (**c**) Complex background.

Due to scale differences, the detection accuracy of medium and large objects is significantly better in comparison with small ones. Therefore, many feature fusion architectures have been developed to reduce small object feature loss and improve overall object detection accuracy. To enhance shallow feature maps' semantic and depth information, the Feature Pyramid Network (FPN) [18] was developed for object detection. Building on the top-down path of FPN, a bottom-up path is added to complement the localization information of deep feature maps in the Path Aggregation Network (PANet) [21]. The feature fusion of deep and shallow layers in FPN and PANet can provide more feature information for object detection. However, these FPN variants have limitations. For example, simply concatenating deep and shallow feature maps does not adequately fuse multi-scale features and may introduce irrelevant information interference. To obtain better results in small object detection, FE-YOLOv5 [19] employs the feature enhancement strategy to enhance the model's spatial perception capability and feature enhancement. QueryDet [20] develops a novel coarse-to-fine cascaded sparse query mechanism. Cao et al. [22] added the mixed attention and the dilated convolution modules in the YOLOv4 network. LMSN [23] proposed multi-scale feature fusion and receptive field enhancement modules to achieve lightweight multi-scale object detection. EFPN [24] developed a feature texture transfer module and a new foreground–background balance loss function. The is-YOLOv5 [25] modified the information path of feature fusion and improved the SPP module. AFPN [26] designed three new attention modules to enhance the model's perception capabilities of foreground and contextual information. Methods to boost the accuracy of small object detection in RS images are still relatively rare. CotYOLOv3 [27] redesigned the residual blocks in the backbone Darknet-53 as Contextual Transformer blocks. FE-CenterNet [28] employs feature aggregation and attention generation structures to detect vehicle objects in RS images.

The baseline RTMDet [17] achieves high accuracy by implementing deep convolution with large convolution kernels and a dynamic soft label assignment strategy. However, the small object detection accuracy improvement of the RTMDet method is not satisfactory. First, multiple convolutional layers in the backbone continuously perform downsampling operations while extracting feature maps. The discriminative features of small objects decrease with decreasing resolution. Second, the deep feature maps extracted by the backbone have low resolution and large receptive fields. These are not appropriate when detecting small objects. Furthermore, small objects may be more negatively affected by the small bounding box perturbations. Inspired by the above small object detection methods, our ERF-RTMDet model introduces two novel modules: the dilated spatial pyramid pooling fast (DSPPF) module and the hybrid dilated attention (HDA) module. Additionally, we adopt the Content-Aware Reassembly of Features (CARAFE) upsampling operator further to boost the small object detection accuracy in RS images. Specifically, we first introduce

high-resolution shallow feature maps. Then, the CARAFE upsampling operator replaces the nearest-neighbor upsampling operator in feature fusion. Finally, we developed the DSPPF and HDA modules to extend the receptive field of small objects. Thus, an improved small object detection approach with the enhanced receptive field, ERF-RTMDet, is proposed to achieve more robust detection capability in RS images. The quantitive detection results of various representative methods on objects of three scale types are compared in Figure 2. Following the definition of the generic MsCOCO dataset [29], in this paper, small objects are defined as objects smaller than $32 \times 32$ pixels, medium objects are defined as ones larger than $32 \times 32$ pixels smaller than $96 \times 96$ pixels, and large objects are defined as ones larger than $96 \times 96$ pixels. The results show that the accuracy of detecting small objects (mAP_s) is significantly worse in comparison with medium (mAP_m) and large ones (mAP_l). The precision in detecting small objects with ERF-RTMDet is 8.8% higher than the most effective comparative method. ERF-RTMDet not only shrinks the distance between the accuracy of detecting small objects and medium and large objects; it can also maintain or even slightly boost the detection effect on medium and large objects.

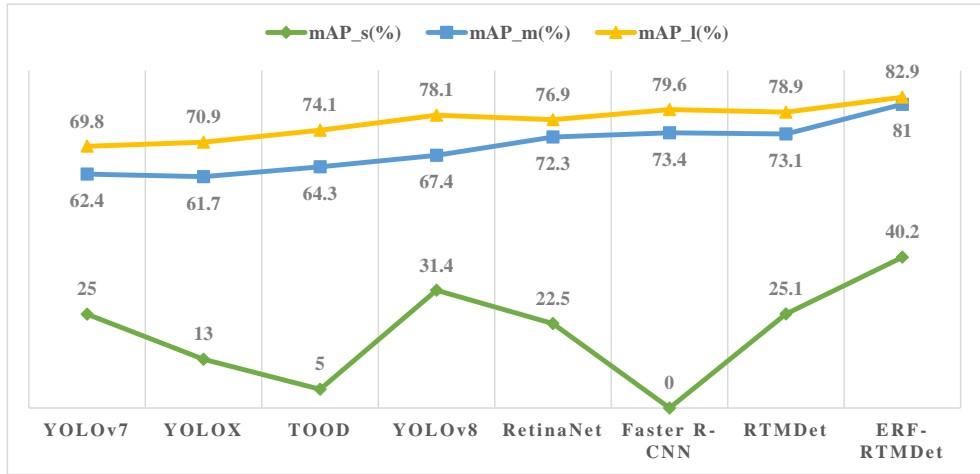

**Figure 2.** Comparison of detection accuracy of different sizes of objects by several methods.

The major contributions of our study are briefly outlined below:

1.  To address the small object detection challenges faced by existing detection methods, the ERF-RTMDet method is proposed to enhance the receptive field and enrich the feature information of small objects.
2.  A dilated spatial pyramid pooling fast module is proposed. The DSPPF module achieves a larger multi-scale receptive field while maintaining resolution. The successive dilated convolutions are employed to extract more contextual information about small objects. The DSPPF module further improves the accuracy through the channel attention module to avoid the interference of background information.
3.  A hybrid dilated attention module is proposed. To obtain more detailed information of small objects, the HDA module uses parallel convolutional layers with different dilation rates to fuse different receptive fields. It is followed by spatial and channel attention modules to extract meaningful information from the fused features.
4.  The CARAFE Module [30] is used in the feature fusion as the upsampling operator rather than the nearest neighbor sampling operator. This allows the upsampled feature maps to contain more information about small objects, leading to the more efficient extraction of object feature information.
5.  The experimental results show that ERF-RTMDet achieves higher detection accuracy on small objects while sustaining or slightly boosting the detection precision on mid-scale and large-scale objects.

## 2. Method

In this section, the general framework of our ERF-RTMDet model is presented. Then, details of the DSPPF module and the HDA module are presented.

### 2.1. Overview of ERF-RTMDet

RS images are usually high resolution. While RS images are used as inputs to the object detection network, they must first be cropped or resized according to different hardware memory and model input sizes. One-stage object detectors can operate faster than two-stage ones. ERF-RTMDet incorporates the improvements made by RTMDet to YOLOX, such as deep convolution with a large kernel size of $5 \times 5$ in the backbone and neck, data augmentation with cache mechanism, and the design of a dynamic soft label assignment strategy. As shown in Figure 3, The backbone CSPNeXt introduced large kernel depth convolution in the cross-stage partial block in CSPDarknet [6], containing four stages, C2–C5. The feature maps output from these four stages are $\{4, 8, 16, 32\}$ times the downsampling rate of the input feature maps, respectively. To boost the capability of detecting small objects, ERF-RTMDet introduces a shallow feature map of higher resolution extracted in stage C2 [31]. The C2 stage's output also contains more detailed information. Instead of the SPPF module, we designed the DSPPF module at the C5 stage, which captures more appropriate receptive fields for small objects. Following YOLOX, the ERF-RTMDet neck combines the characteristics of FPN and PAN. The feature maps generated by the concatenation operation are first fed into the CSP block before being passed to the convolution module. The HDA module is added after the 32-times-downsampled feature map output from the neck to enhance the representative capability of the low-resolution feature maps. Additionally, the convolution module replaces the regular convolution with a dilated convolution [32] with an expansion rate of two when extracting the low-resolution feature maps in the backbone (C4 and C5) and neck (P4 and P5). Such a modification expands the receptive field of the deep feature maps and preserves more detail, allowing for better object detection. Finally, the four-layer feature maps are fused and fed to the following detection heads for classifying and locating. The convolution modules of the classification and regression detection heads share their parameters, respectively.

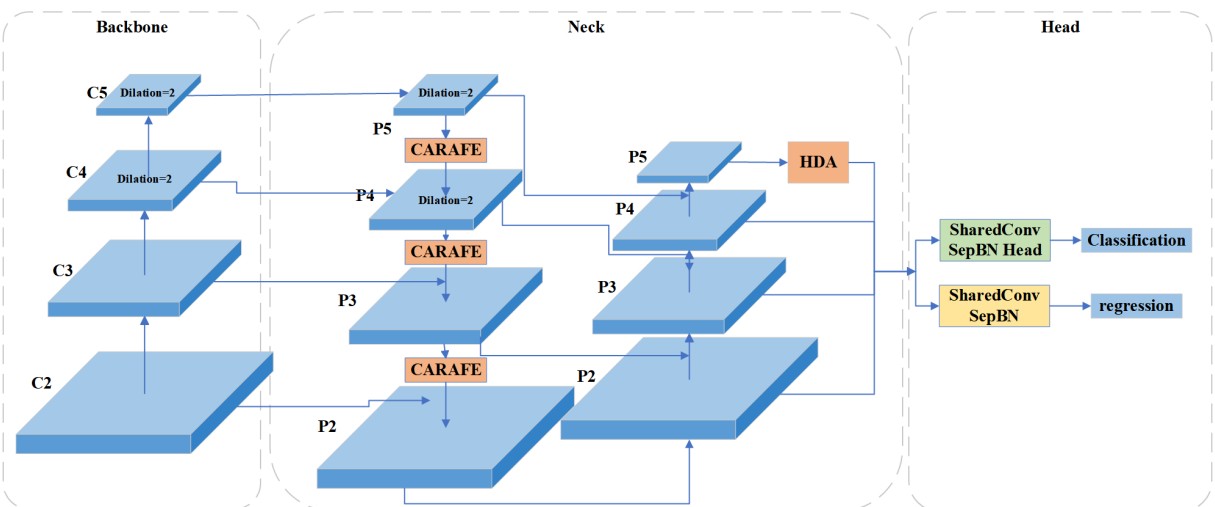

**Figure 3.** An Overview of Our ERF-RTMDet. CARAFE Module: Content-Aware Reassembly of Features Module. HDA Module: Hybrid Dilated Attention Module.

### 2.2. CARAFE Upsampling Operator

In the top-down path of the neck feature fusion network, feature maps of different resolutions are generally upsampled to the same resolution before being concatenated. Many current small object detection methods in the RS datasets rely on the original nearest-neighbor interpolation in the FPN, and others employ deconvolution [33] or sub-pixel

convolution [27] instead. The nearest-neighbor interpolation method fills the feature map using adjacent pixels and only considers sub-pixel neighborhoods. Also, when upsampling with the nearest-neighbor interpolation, much of the feature information of small objects is lost due to background interference. Deconvolution scales up the size by padding the feature map with zeros. Much invalid information is introduced while improving the resolution. It is not ideal for small objects in RS images. Sub-pixel convolution [34] first extends the channel dimension of the feature map. After that, the multiple of the channel dimension extension is transferred to the specific values for feature map dimensions by the PixelShuffle operation. CARAFE is proposed as a lightweight and efficient upsampling operator with several advantages over the above operators. First, CARAFE has a larger receptive field compared to sub-pixel neighborhoods. This allows for a more comprehensive use of contextual information. Second, unlike deconvolution, which employs a fixed kernel, CARAFE produces adaptive kernels in an on-the-fly content-aware approach. Specifically, CARAFE predicts the upsampled kernels by convolutional layers. Different kernels are applied to combine semantic information at various positions in the feature map, achieving the final upsampling result. Moreover, CARAFE can be seamlessly integrated into various architectures that contain upsampling operators to achieve lightweight and operationally fast feature upsampling. Therefore, in the neck feature fusion network, ERF-RTMDet uses the plug-and-play CARAFE upsampling operator instead of nearest-neighbor interpolation to generate upsampled feature maps, which can more accurately represent the object shape.

### 2.3. Dilated Spatial Pyramid Pooling Fast Module

As the feature maps extracted by the backbone network become deeper layer by layer, the receptive fields of the feature maps gradually expand. To fuse receptive fields of different scales and obtain richer contextual information, YOLOv4 introduces the Spatial Pyramid Pooling (SPP) module [35]. Furthermore, the SPPF module in YOLOv5 replaces the parallel max-pooling layer with a sequential method to boost the running speed of the module. However, the ordinary convolution operation outputs a fixed receptive field and extracts a limited feature region. The max-pooling layer reduces the resolution of the feature map. It also suffers from the drawback of inaccessible spatial localization information. To address this issue, ASPP [36] employs several convolutions with different dilation rates parallelly to fuse multi-scale feature information. Inspired by the above methods, the proposed DSPPF module substitutes the SPPF module in the backbone C5 stage to enhance the small object detection accuracy. Figure 4 provides an illustration of our ERF-RTMDet's C5 stage.

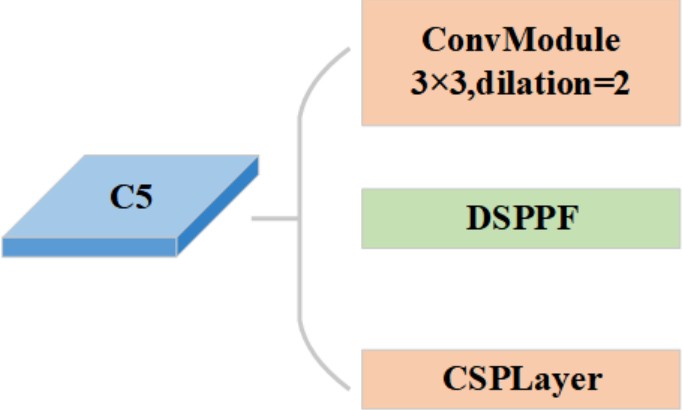

**Figure 4.** Illustration of ERF-RTMDet backbone C5 stage. DSPPF Module: Dilated Spatial Pyramid Pooling Fast Module. CSPLayer: Cross-Stage Partial Layer.

A detailed comparison between the framework of the SPPF module and the DSPPF module is presented in Figure 5.

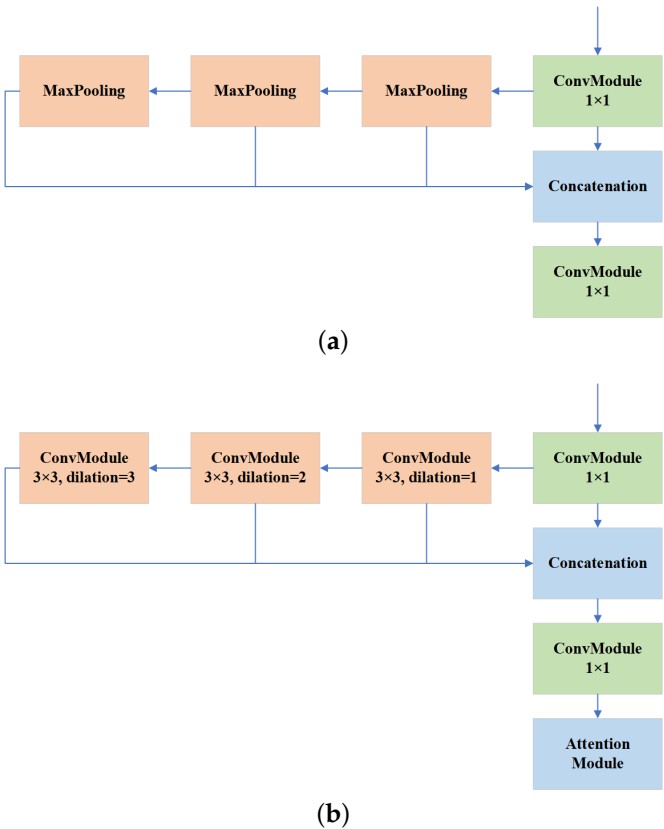

**Figure 5.** Comparison of the framework between the Spatial Pyramid Pooling Fast(SPPF) module and the Dilated Spatial Pyramid Pooling Fast(DSPPF) module. (**a**) SPPF module. (**b**) DSPPF module.

The DSPPF module consists of sequential convolution modules with different dilation rates and an attention module. Specifically, the $1 \times 1$ convolution $Conv_{1 \times 1}$ reduces the number of input channels, thereby reducing the number of parameters and improving operational efficiency. The convolution kernel of dilated convolution is discontinuous. Not all pixels are used in the computation. Continuous dilated convolution can cause a gridding effect if the dilation rates are not set appropriately. The receptive field of feature maps is obtained grid-like, which loses locally relevant information and may also introduce irrelevant distant information. The solution is to make the final output of a series of convolution operations have a square receptive field without holes or edge loss [37]. The corresponding equation is given as follows:

$$M_i = max[M_{i+1} - 2r_i, M_{i+1} - 2(M_{i+1} - r_i), r_i], \tag{1}$$

where $N$, $K$, and $r$ represent the number of convolution layers, convolution kernel size, and convolution layer dilation rate, respectively; $M_i$ represents the maximum distance between two nonzero values. The equation satisfies both $M_N = r_N$ and $M_2 \leq K$. For the proposed DSPPF module, the convolution kernel size K is three. Meanwhile, employing ordinary convolution in the first layer can avoid the underlying information loss. Therefore, the dilation rates of the three convolutional layers in the DSPPF module are $r = 1, 2, 3$. After $Conv_{1 \times 1}$, an ordinary convolution with $3 \times 3$ kernel size $Conv_{3 \times 3, r=1}$ is applied first. Then, two $3 \times 3$ convolution modules with dilation rates of two and three $Conv_{3 \times 3, r=2}$ and $Conv_{3 \times 3, r=3}$ are applied to extract feature maps of different receptive fields. Four different receptive fields' feature maps are concatenated and reduced in dimension. Finally, the Squeeze-and-Excitation (SE) attention module [38] provides the output. For the input feature map $f_{in}$, the specific procedure of the DSPPF module is denoted by the following equations:

$$f_{out1} = Conv_{1 \times 1}(f_{in}), \tag{2}$$

$$f_{out2} = Conv_{3\times3, r=1}(f_{out1}), \tag{3}$$

$$f_{out3} = Conv_{3\times3, r=2}(f_{out2}), \tag{4}$$

$$f_{out4} = Conv_{3\times3, r=3}(f_{out3}), \tag{5}$$

$$f_{out} = SEModule(Conv_{1\times1}(Concat(f_{out1}, f_{out2}, f_{out3}, f_{out4}))). \tag{6}$$

The SE attention module is added after the concatenated features, as shown in Figure 6. First, the information is summarized and compressed by the global average pooling layer to learn the relevance between channel feature maps. Then, two $1 \times 1$ convolution modules are implemented instead of the original full convolution layer to excite the channel weights. The attention weights acquired in the process will then multiply with the original input to produce the ultimate output.

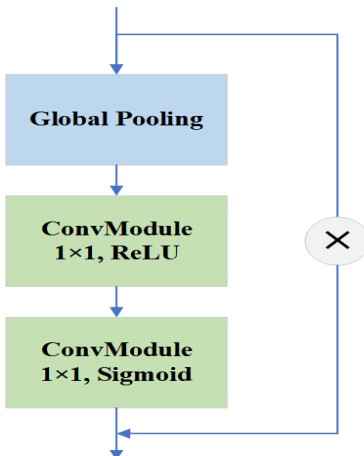

**Figure 6.** The framework of the Squeeze-and-Excitation (SE) attention module.

*2.4. Hybrid Dilated Attention Module*

After the neck feature fusion, the output feature maps are $\{4, 8, 16, 32\}$ times the downsampled size of the input, respectively. The 32-times-downsampled deep feature maps have a small resolution, which increases the challenge of object detection. The HDA module is added after the lowest-resolution feature map. This enhances its ability to interpret contextual information. The framework of the HDA module is described in Figure 7. Similarly, the $1 \times 1$ convolution operation first decreases the dimensionality of the channels and the parameters of the module. The reduced dimensionality output is fed into three convolutional branches and one residual attention branch in parallel. Considering the kernel size of 3 and Equation (1), the dilation rates of the three parallel convolutions are set to 1, 2, and 4. The residual attention branch comprises two parts: channel attention and spatial attention. Specifically, the channel attention applies the ECA module, which can obtain relatively good accuracy without slowing the speed. For a given input $fm_{in}$, the specific procedures of spatial attention are shown below,

$$fm_{out} = Conv'_{1\times1}(Conv_{3\times3, r=2}(Conv_{3\times3, r=2}(Conv_{1\times1}(fm_{in})))), \tag{7}$$

where the $Conv_{3\times3, r=2}$ and $Conv_{1\times1}$ denote convolution modules containing a convolution layer, a batch normalization layer, and a ReLU activation layer. $Conv'_{1\times1}$ denotes a convolution layer with only one convolution layer. Finally, these four branches aggregate contextual information with different receptive fields. They are concatenated and sent to the detection head for RS image object detection.

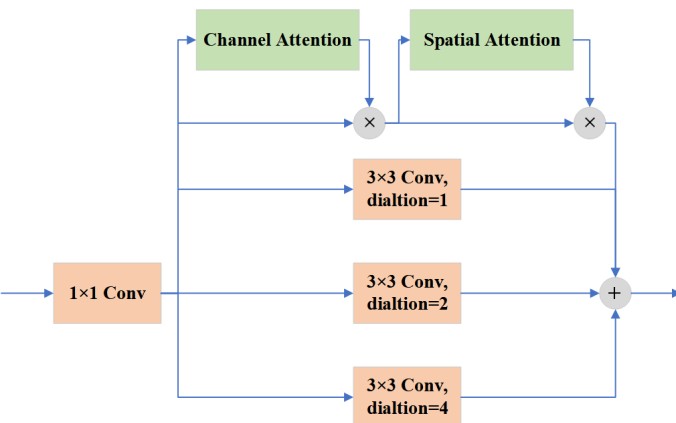

**Figure 7.** The framework of Hybrid Dilated Attention module.

### 2.5. Loss Function

The loss function $L_{det}$ of ERF-RTMDet contains the Quality Focal Loss [39] $L_{cls}$ used for classification and the GIoU loss [40] $L_{reg}$ used for bounding box regression. Focal Loss [11] is usually used as the classification loss for the detection head of the one-stage methods. Quality Focal Loss extends Focal Loss to enable the use of continuous labels that combine classification and localization quality. $L_{cls}$ is defined as follows:

$$L_{cls} = -|y - \sigma|^{\beta}((1 - y)log(1 - \sigma) + ylog(\sigma)), \tag{8}$$

where $y$ denotes the continuous label obtained from the IoU of the bounding box and the ground truth; $\sigma$ denotes the predicted value; $\beta$ denotes the hyperparameter of the dynamic scale factor. The GIoU loss can be optimized while the overlapping area of two boxes is zero, with good stability. $L_{reg}$ is defined as follows:

$$L_{reg} = 1 - \left(\frac{|Gt \cap P|}{|Gt \cup P|} - \frac{|C - Gt \cup P|}{|C|}\right), \tag{9}$$

where $Gt$, $P$, and $C$ denote the ground truth box, the prediction box, and the smallest enclosing convex box of these two boxes, respectively. The total loss function $L_{det}$ of ERF-RTMDet is denoted as,

$$L_{det} = \lambda_1 L_{cls} + \lambda_2 L_{reg}, \tag{10}$$

where $\lambda_1$ and $\lambda_2$ are weights of classification loss and regression loss, set to one and two by default, respectively.

## 3. Result

### 3.1. Datasets and Evaluation Metrics

The NWPU VHR-10 dataset and MAR20 dataset are utilized in this study to evaluate the proposed ERF-RTMDet and other comparative methods.

**NWPU VHR-10 Dataset:** The NWPU VHR-10 dataset [41] is a public geospatial object detection dataset. It consists of 800 very-high-resolution RS images gathered through Google Earth and Vaihingen. Images are $500 \times 500 \sim 1100 \times 1100$ pixels in size. NWPU VHR-10 dataset contains 3651 instances of 10 types of RS objects. The average size of the objects in these ten categories is about 6.4% of the image size. In evaluating various methods, the horizontal bounding box annotations provided by the NWPU VHR-10 dataset are used.

**MAR20 Dateset:** The MAR20 dataset [42] is the most massive RS image military aircraft object recognition dataset available. It consists of 3842 high-resolution images collected through Google Earth at 60 military airports in the United States, Russia, and other countries. The size of images is mostly $800 \times 800$ pixels. The MAR20 dataset contains 22,341 instances of 20 military aircraft categories. We used category labels A1~A20 to represent

these 20 aircraft types, respectively. The average size of the objects in these 20 categories is about 2.2% of the image size. MAR20 dataset is a fine-grained classification of military aircraft. Different aircraft types may have similar characteristics, so the inter-class variation is small. External factors such as weather and occlusion during RS image acquisition make the dataset's intra-class variation large. Precisely detecting and identifying the correct category of aircraft instances is arduous. The horizontal bounding box annotations provided by the MAR20 dataset were used in the evaluation of the different methods.

**Evaluation Metrics:** In order to evaluate the detection accuracy of various methods for RS objects, we adopt the COCO evaluation metrics. The metrics include: mAP_50, the average of all categories with an IoU threshold of 0.5; mAP, the average of mAP for different IoU thresholds from 0.5 to 0.95; mAP_s, mAP for small objects with area less than $32 \times 32$; mAP_m, mAP for medium objects with area between $32 \times 32$ and $96 \times 96$; and mAP_l, mAP for large objects with area larger than $96 \times 96$.

### 3.2. Implementation Details

All experiments in this study were conducted on a single NVIDIA GeForce RTX 3080 GPU using PyTorch. Due to the limited hardware device computation resources, all images were cropped to $320 \times 320$ pixels during training. We divided the cropped NWPU VHR-10 dataset and MAR20 dataset in the ratio of 6:2:2. In the NWPU VHR-10 dataset, 2088 images were used as the training set, 522 images as the validation set, and 653 images as the test set. In the MAR20 dataset, 10,635 images were used as the training set, 2659 images were used as the validation set, and 615 original images were used as the test set. For all datasets and comparative methods, the total training epochs and batch sizes were set to 200 and 4. In the testing stage, we use the model with the highest mAP within the 200 training epochs as the evaluation model. All optimizers for the comparative methods were set to AdamW. The learning rate had an initial value of $lr = 0.00005$, following a cosine decay schedule from the 100~200 epochs with a weight decay of $0.05lr$.

### 3.3. Comparison Experiments

To validate the effectiveness of ERF-RTMDet in RS image object detection, we compared it with several representative methods on the NWPU VHR-10 and MAR20 datasets. Methods for comparison include Faster R-CNN [2], RetinaNet [11], YOLOX [16], TOOD [43], etc. Quantitative comparison results on the NWPU VHR-10 and MAR20 datasets are shown in Tables 1 and 2, with the best results in bold. The proposed ERF-RTMDet achieves the best mAP and mAP_s on both NWPU VHR-10 and MAR20 datasets.

**Results on NWPU VHR-10 dataset:** Compared with representative one-stage and two-stage object detection methods, the proposed ERF-RTMDet achieves more precise object detection results. Over the strong baseline RTMDet, ERF-RTMDet obtained an improvement of 1.9% mAP and achieved 56.8%, 93.4%, and 52.2% for mAP, mAP_50, and mAP_s, respectively. ABNet [44] is a framework specifically designed for multi-scale object detection in RS images. Referring to the reported result in the ABNet paper, ERF-RTMDet obtains an improvement of 2.68% mAP_50. Quantitative comparison results show that ERF-RTMDet can achieve higher detection accuracy on small objects while maintaining or slightly improving the detection accuracy on mid-scale and large-scale objects.

**Results on MAR20 dataset:** Similarly, the proposed ERF-RTMDet achieved the best object detection results in comparative methods. Compared with the strong baseline RTMDet, ERF-RTMDet obtains an improvement of 5.2% mAP and achieves 81.5%, 99.7%, and 40.2% for mAP, mAP_50, and mAP_s, respectively. Quantitative comparison results show that ERF-RTMDet achieves optimal or sub-optimal detection accuracy on all categories in the MAR20 dataset.

**Table 1.** Comparison of the detection accuracy of different methods on the NWPU VHR-10 dataset. The abbreviations stand for storage tank (ST), baseball diamond (BD), basketball court (BC), tennis court (TC), and ground track field (GTF). The AP(%) was taken as the metric for each category. Bold print highlights the best results. Sub-optimal results are underlined.

| Category | Method | | | | | | | |
|---|---|---|---|---|---|---|---|---|
| | **CornerNet** | **RetinaNet** | **YOLOX** | **YOLOv7** | **Faster R-CNN** | **RTMDet** | **YOLOv8** | **ERF-RTMDet** |
| Airplane | 54.2 | 58.4 | 62.1 | 59.4 | 63.9 | <u>64.3</u> | **66.8** | 63.8 |
| Ship | 47.6 | 46.8 | 52.0 | 53.3 | 54.7 | **60.2** | 59.4 | <u>59.8</u> |
| ST | 28.0 | 53.1 | 45.9 | 44.3 | <u>60.3</u> | 55.3 | 56.3 | **60.9** |
| BD | 54.9 | 55.2 | 60.5 | 61.9 | 58.5 | **68.3** | 64.2 | <u>65.9</u> |
| TC | 28.6 | 49.4 | 55.9 | 55.5 | **61.3** | **61.3** | <u>58.9</u> | 58.1 |
| BC | 37.6 | 43.2 | 57.2 | 55.6 | <u>64.1</u> | 62.1 | 63.1 | **67.4** |
| GTF | <u>45.3</u> | 23.5 | 24.7 | 32.6 | 35.6 | 41.0 | 33.8 | **45.6** |
| Habor | 27.1 | 32.5 | 47.4 | **51.0** | 41.3 | 48.6 | <u>50.8</u> | 49.8 |
| Bridge | 20.9 | 17.1 | 30.4 | <u>39.6</u> | 33.5 | 31.8 | **40.9** | 37.7 |
| Vehicle | 30.7 | 45.5 | 51.6 | 49.9 | <u>59.6</u> | 56.0 | **62.7** | 58.4 |
| **mAP_s** | 12.5 | 31.4 | 33.5 | 16.2 | <u>35.5</u> | 26.6 | 32.9 | **52.0** |
| **mAP_m** | 45.7 | 48.7 | 52.1 | 53.0 | **59.1** | 55.7 | <u>58.8</u> | 58.6 |
| **mAP_l** | 38.0 | 42.0 | 51.9 | 54.8 | 50.0 | **61.4** | <u>58.0</u> | 55.3 |
| **mAP_50** | 57.4 | 77.2 | 83.1 | 92.8 | 85.5 | **93.5** | 91.6 | <u>93.4</u> |
| **mAP** | 37.5 | 42.5 | 48.8 | 50.3 | 53.3 | 54.9 | <u>55.7</u> | **56.8** |

**Table 2.** Comparison of the detection accuracy of different methods on the MAR20 dataset. The AP(%) was taken as the metric for each category. Bold print highlights the best results. Sub-optimal results are underlined.

| Category | Method | | | | | | | |
|---|---|---|---|---|---|---|---|---|
| | **YOLOv7** | **YOLOX** | **TOOD** | **YOLOv8** | **RetinaNet** | **Faster R-CNN** | **RTMDet** | **ERF-RTMDet** |
| A1 | 62.7 | 66.8 | 67.1 | 71.5 | 74.2 | 74.1 | <u>75.5</u> | **78.2** |
| A2 | 68.8 | 72.7 | 75.8 | 76.0 | <u>78.4</u> | 78.2 | 75.6 | **81.2** |
| A3 | 67.6 | 60.0 | 69.6 | 72.6 | 75.4 | 75.2 | 76.5 | **80.1** |
| A4 | 74.0 | 77.2 | 67.6 | 79.2 | 81.8 | <u>82.8</u> | <u>82.8</u> | **88.1** |
| A5 | 58.8 | 56.1 | 60.4 | 64.5 | <u>70.0</u> | 69.5 | 67.7 | **75.2** |
| A6 | 69.9 | 69.2 | 77.8 | 73.8 | 77.2 | <u>79.9</u> | 79.2 | **84.6** |
| A7 | 70.8 | 71.0 | 74.1 | 79.3 | 79.3 | **81.4** | 80.6 | <u>81.1</u> |
| A8 | 75.1 | 73.5 | 77.3 | 79.1 | 78.1 | 80.3 | <u>80.9</u> | **85.0** |
| A9 | 71.2 | 70.7 | 69.6 | 75.6 | <u>79.3</u> | 78.8 | 76.9 | **83.9** |
| A10 | 72.5 | 73.3 | 73.4 | 75.5 | 76.3 | 76.9 | <u>77.0</u> | **80.2** |
| A11 | 70.7 | 65.9 | 68.7 | 76.2 | 69.7 | <u>81.4</u> | 76.8 | **82.1** |
| A12 | 60.6 | 70.1 | 70.5 | 72.2 | 72.0 | **80.3** | 74.6 | <u>77.7</u> |
| A13 | 63.7 | 63.2 | 66.9 | 67.6 | 71.0 | <u>71.4</u> | 69.2 | **78.6** |
| A14 | 75.3 | 72.5 | 75.5 | 79.3 | 79.5 | 79.0 | <u>80.0</u> | **82.3** |
| A15 | 52.2 | 46.1 | 47.4 | 59.0 | 65.5 | 65.5 | <u>65.7</u> | **80.8** |
| A16 | 71.6 | 75.1 | 71.5 | 77.9 | 75.8 | 76.8 | <u>78.2</u> | **81.1** |
| A17 | 69.4 | 72.0 | 78.6 | 77.2 | <u>81.1</u> | 80.8 | 80.9 | **86.5** |
| A18 | 75.1 | 77.5 | 72.3 | 81.8 | 83.0 | <u>84.3</u> | 83.6 | **85.5** |
| A19 | 54.9 | 59.0 | 62.2 | 65.1 | <u>75.0</u> | 74.7 | 72.4 | **81.8** |
| A20 | 56.2 | 58.2 | 72.1 | 67.9 | <u>75.0</u> | <u>75.0</u> | 71.3 | **75.3** |
| **mAP_s** | 25.0 | 13.0 | 5.0 | <u>31.4</u> | 22.5 | 0.0 | 25.1 | **40.2** |
| **mAP_m** | 62.4 | 61.7 | 64.3 | 67.4 | 72.3 | <u>73.4</u> | 73.1 | **81.0** |
| **mAP_l** | 69.8 | 70.9 | 74.1 | 78.1 | 76.9 | <u>79.6</u> | 78.9 | **82.9** |
| **mAP_50** | 95.8 | 92.7 | 90.0 | 95.9 | 96.5 | 97.3 | <u>98.1</u> | **99.7** |
| **mAP** | 67.1 | 67.5 | 69.9 | 73.6 | 75.9 | <u>76.9</u> | 76.3 | **81.5** |

**Small Object Detection Results:** Observing the mAP_s metric on the NWPU VHR-10 dataset, ERF-RTMDet improves by 20.6%, 18.5%, and 25.4% over RetinaNet, YOLOX, and RTMDet, respectively. The small object detection accuracies achieved by ERF-RTMDet on the MAR20 dataset are enhanced by 17.7%, 27.2%, and 15.1% over RetinaNet, YOLOX, and RTMDet, respectively. Examples of feature visualization results for several models are presented in Figure 8. The RetinaNet model did not effectively capture all objects. The YOLOX and RTMDet models focused more on a single object of interest. In comparison,

the ERF-RTMDet model can focus on more objects and has a larger effective receptive field. The comparison of quantitative and visualization results proves the effectiveness of the proposed ERF-RTMDet. ERF-RTMDet is able to obtain a more robust ability to detect small objects while sustaining or slightly boosting the detection effect on mid-scale and large-scale objects.

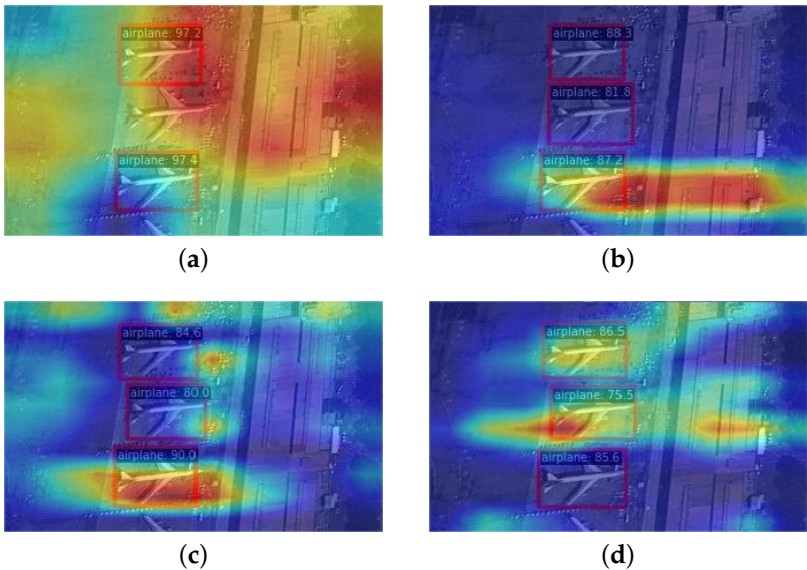

**Figure 8.** Feature visualization results of different methods. (**a**) RetinaNet. (**b**) YOLOX. (**c**) RTMDet. (**d**) Our ERF-RTMDet.

Overall, the proposed ERF-RTMDet obtains the best object detection accuracy on the NWPU VHR-10 and MAR20 datasets. We select two comparative methods with better accuracy in Tables 1 and 2 to compare with ERF-RTMDet qualitatively. Several visualization results on the NWPU VHR-10 and MAR20 datasets are shown in Figures 9 and 10, respectively. Comparing the fourth row in Figure 9 and the fourth row in Figure 10, the proposed ERF-RTMDet has higher accuracy and lower miss detection rate in detecting densely arranged RS objects. While detecting small instances in RS images, such as the ship in the fifth row in Figure 9 and the A20 type in the fifth row in Figure 10, ERF-RFMDet also achieves the most advanced accuracy among several comparative methods. Both qualitative and quantitative comparison experiments demonstrate that ERF-RTMDet can effectively address the challenges of small variation between classes and large variation within classes in the MAR20 dataset. Thus, our ERF-RTMDet can enhance the small object detection accuracy while sustaining or slightly boosting the detection accuracy on mid-scale and large-scale objects.

*3.4. Ablation Study*

To validate the effectiveness of our modifications to the baseline RTMDet, ablation experiments were performed on the MAR20 dataset for different modules. The tables below show the six improvements we added, respectively. Where the baseline indicates the RTMDet method, ∘ and × indicate with and without adding the corresponding improvements. Table 3 verifies that introducing the shallow and large feature map of the C2 stage can capture more details and localization information of objects. The introduction of the C2 stage boosts the overall detection accuracy by 2.3% mAP and 1.6% mAP_50, respectively, although a slight decrease of 0.6% mAP_s is observed. Adopting CARAFE instead of nearest-neighbor interpolation as the upsampling operator in the feature fusion procedure can further improve the overall detection accuracy. Small object detection accuracy is also boosted at the same time.

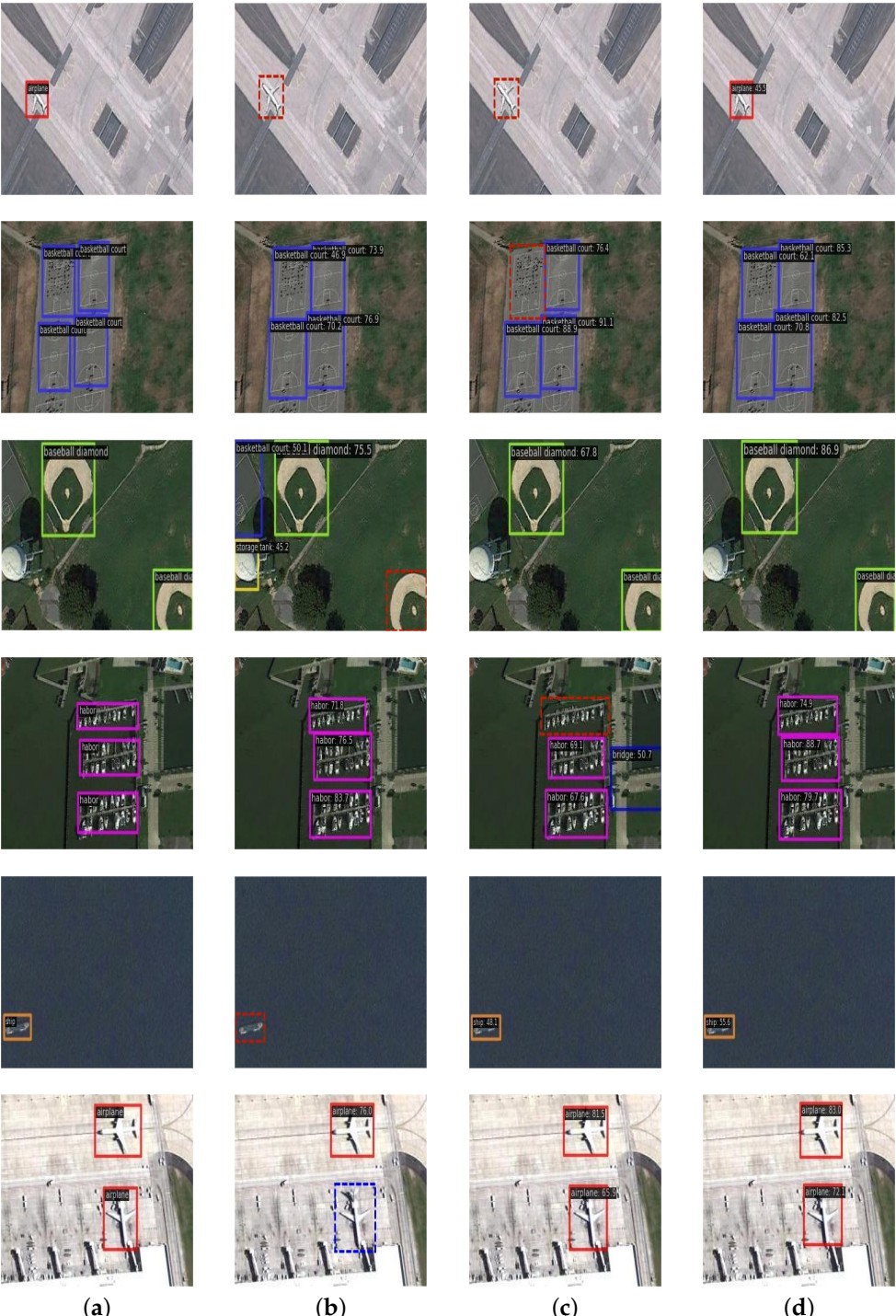

**Figure 9.** Visual results obtained by various methods on the NWPU VHR-10 dataset. (**a**) Ground truth. (**b**) RTMDet. (**c**) YOLOv8. (**d**) Our ERF-RTMDet. Solid boxes of different colors represent different categories of the objects. The miss detections are shown in bold dash boxes.

Table 4 demonstrates the effectiveness of applying dilated convolution instead of regular convolution in backbone and neck networks. The overall and small object detection accuracy has been enhanced.

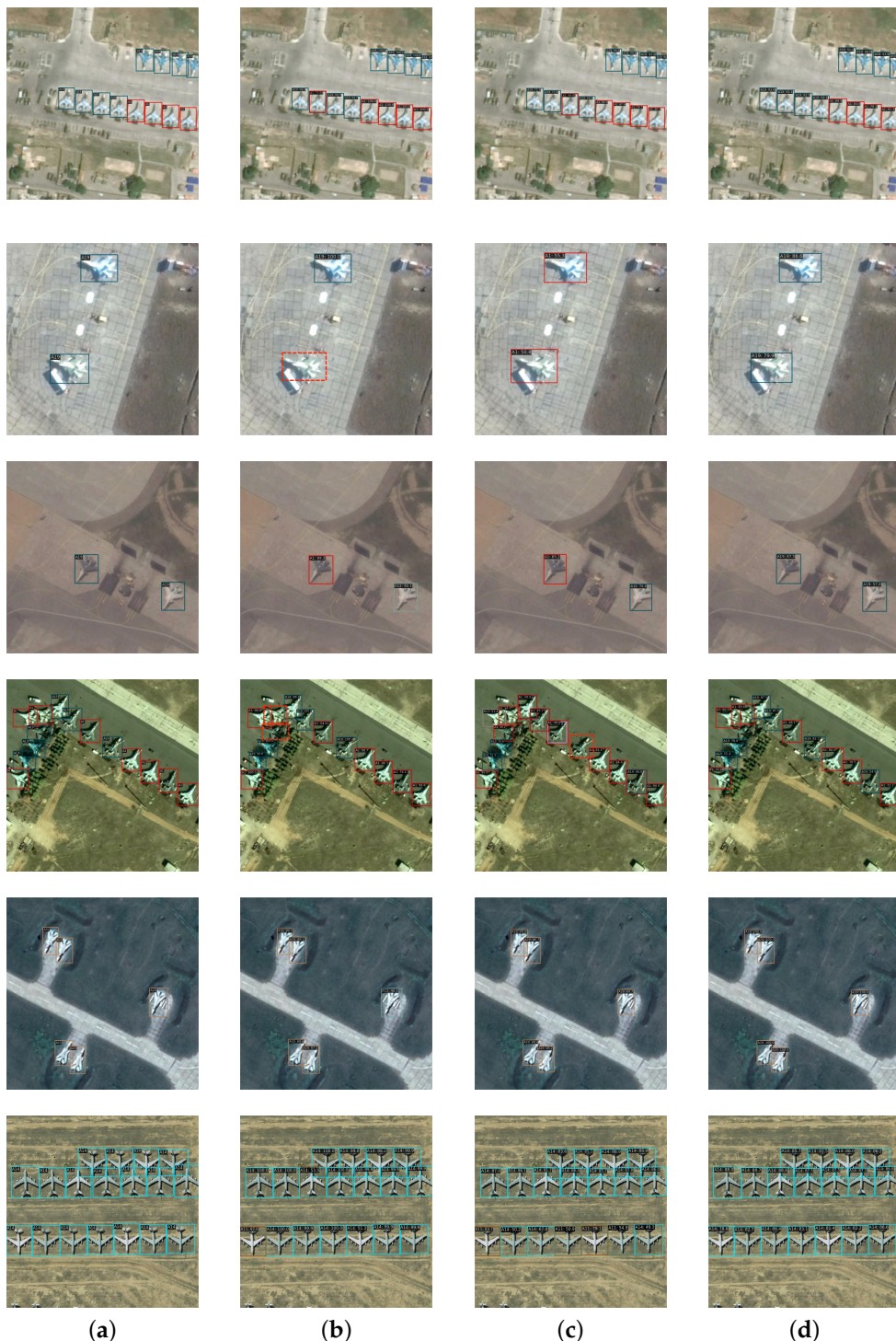

**Figure 10.** Visual results obtained by various methods on the MAR20 dataset. (**a**) Ground truth. (**b**) Faster R-CNN. (**c**) RTMDet. (**d**) Our ERF-RTMDet. The miss detections are shown in red bold dash boxes.

Table 5 validates the impact of the HDA module and DSPPF module on the detection accuracy. The HDA and DSPPF modules obtain more contextual information using continuous dilated convolution. Meanwhile, the attention mechanism introduced allows the model to focus more on valuable information and avoid the interference of background information.

**Table 3.** Ablation experiments of whether to introduce the CARAFE upsampling operator and C2 stage. ○ and × indicate with and without adding the corresponding improvements.

|  | C2 | CARAFE | mAP | mAP_50 | mAP_s |
| --- | --- | --- | --- | --- | --- |
| Baseline | × | × | 73.4 | 97.2 | 40.5 |
| Baseline | ○ | × | 75.7 | 98.8 | 39.9 |
| Baseline | ○ | ○ | 75.7 | 99.2 | 40.1 |

**Table 4.** Ablation experiments of whether to introduce dilated convolutions in the backbone and neck. ○ and × indicate with and without adding the corresponding improvements.

|  | Dilated Neck | Dilated Backbone | mAP | mAP_50 | mAP_s |
| --- | --- | --- | --- | --- | --- |
| Baseline | × | × | 73.4 | 97.2 | 40.5 |
| Baseline | ○ | × | 75.7 | 99.4 | 41.3 |
| Baseline | ○ | ○ | 76.0 | 99.3 | 42.1 |

**Table 5.** Ablation experiments of whether to add HDA module and DSPPF module. The best results are shown in bold. ○ and × indicate with and without adding the corresponding improvements.

|  | HDA | DSPPF | mAP | mAP_50 | mAP_s |
| --- | --- | --- | --- | --- | --- |
| Baseline | × | × | 73.4 | 97.2 | 40.5 |
| Baseline | ○ | × | 75.8 | 98.9 | 42.9 |
| Baseline | ○ | ○ | **76.2** | **99.3** | **44.1** |

## 4. Discussion

Existing methods have made some attempts to boost the precision of detecting small objects. For example, multi-scale feature fusion is conducted by using high-resolution features from lower layers. However, the semantic information of shallow feature maps needs to be more reliable and requires further processing. Attention mechanisms are introduced to highlight the objects of interest in the feature maps and suppress the background noise. Some other approaches use super-resolution networks to preprocess images to enhance the representation of small target feature maps. However, super-resolution networks are often complex and computationally expensive. Moreover, most methods focus on detecting small objects in natural images. Methods that specifically study the detection of small objects in RS images are less available. Therefore, this paper proposes the DSPPF and HDA modules to extract feature maps with richer contextual information. Moreover, the dilated convolution and the more efficient upsampling operator CARAFE are incorporated to enlarge the receptive field of feature maps as well as to learn representations more conducive to localization. Both comparison and ablation experiments demonstrate the efficiency of the proposed ERF-RTMDet.

### 4.1. DSPPF Module

Following the sequential dilated convolution, the DSPPF module introduces the SE attention mechanism. Besides the SE module, some other attention mechanisms have been introduced. For example, the Convolutional Block Attention Module (CBAM) [45] has two branches to learn channel and spatial attention. So, it has a more complex structure and low operational efficiency. The Efficient Channel Attention (ECA) module [46] uses one-dimensional convolution to implement a local cross-channel interaction strategy. These three attention mechanisms are compared in experiments, as shown in Table 6. The introduction of the SE attention mechanism in the DSPPF module yields the best effect, as shown in the experimental results.

**Table 6.** Comparison of different attention mechanisms in the DSPPF module. The best results are shown in bold.

| Attention Module | mAP_s | mAP_50 | mAP |
|---|---|---|---|
| None | 44.0 | 99.3 | 76.1 |
| CBAM | 43.6 | 98.3 | 75.1 |
| ECA | 41.3 | 99.3 | 76.2 |
| SE | **48.9** | **99.4** | **76.2** |

### 4.2. HDA Module

The HDA module employs parallel convolutions with different dilation rates to acquire the multi-scale information necessary to detect small objects without diminishing the receptive field. One residual attention branch is included in the HDA module. We experimentally compared the effect of attention modules added at different locations on accuracy, as shown in Table 7. Experimental results demonstrate that the effect of the attention module added in the residual branch is superior to that after the concatenation operation.

**Table 7.** Comparison of attention mechanisms added at different locations in the HDA module. The best results are shown in bold.

| Attention Module's Location | mAP_s | mAP_50 | mAP |
|---|---|---|---|
| After concatenation | 40.0 | 99.3 | 78.5 |
| In residual branch | **40.2** | **99.7** | **81.5** |

### 4.3. Future Work

In future work, we intend to enhance our ERF-RTMDet from three perspectives. Firstly, due to the limited memory, the input images have been resized to $320 \times 320$ pixels. We will attempt to access the computational cluster to evaluate the detection results of different models on larger input images (e.g., $1280 \times 1280$ pixels) using larger hardware memory. Secondly, the NWPU VHR-10 and MAR20 remote sensing datasets utilized in this paper contain 10 and 20 categories of objects, respectively. We will attempt to explore whether there are available RS datasets suitable for testing edge cases (e.g., containing only one class or a large number of classes). Based on this, we will evaluate the detection performance of our ERF-RTMDet on such datasets and verify its scalability and versatility. Finally, the NWPU VHR-10 and MAR20 remote sensing datasets used in this paper are high-resolution remote sensing image datasets. We will attempt to collate and simulate new datasets that contain moving objects. Based on the new dataset, the detection performance of our ERF-RTMDet on moving objects will be detected. Also, we will further explore new strategies to improve the detection performance of our ERF-RTMDet on low- and medium-resolution remote sensing image datasets.

### 5. Conclusions

A small object detection method with the enhanced receptive field, ERF-RTMDet, is proposed by us to obtain more robust detection capability on small objects in RS images. Specifically, the DSPPF module is designed to obtain more contextual information about small objects and avoid the interference of background information. Additionally, we substituted the nearest neighbor interpolation operator with the CARAFE operator in our feature fusion network for more effective upsampling. The HDA module is employed to augment the receptive field of features and enhance the features of objects. Comprehensive experimental results validate the efficacy of the ERF-RTMDet on the publicly available datasets MAR20 and NWPU VHR-10. Compared with other most representative models, our ERF-RTMDet can both obtain better detection results on small objects and maintain or slightly boost the detection effect on mid-scale and large-scale objects.

**Author Contributions:** Conceptualization, H.Z.; methodology, S.L.; software, S.L.; validation, S.L. and H.Z.; formal analysis, S.L.; investigation, S.L.; resources, S.L.; data curation, S.L.; writing—original draft preparation, S.L.; writing—review and editing, S.L. and H.Z.; visualization, S.L. and S.H.; supervision, X.C., Y.H., M.L. and Y.Z.; project administration, S.L. All authors have read and agreed to the published version of the manuscript.

**Funding:** This work was supported by the Natural Science Foundation of China under grant 62071474.

**Data Availability Statement:** The MAR20 dataset used in this study (accessed on 16 March 2023) is accessible from https://gcheng-nwpu.github.io/. The NWPU dataset used in this study (accessed on 24 November 2022) is accessible from http://pan.baidu.com/s/1hqwzXeG.

**Conflicts of Interest:** The authors declare no conflict of interest.

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
