# Peer review of "ERF-RTMDet: An Improved Small Object Detection Method in Remote Sensing Images"

_remotesensing, doi:10.3390/rs15235575_

Round 1
Reviewer 1 Report
Comments and Suggestions for Authors
The manuscript discussed a detection method for small objects. The used technologies included deep learning based YOLO algorithms, the Pyramid Pooling, upsampling, attention mechanism, etc. The following suggestions are provided for authors’ consideration:
1. The contributions of the manuscript could be clarified further. For example, it is claimed by the authors that the ERF-RTMDet method is proposed to enhance the receptive field and enrich the feature information of small objects. However, it seems that the proposed method is based on YOLOX, and the other major contributions came from CSPNeXt backbone, CSPDarknet, etc. So it is better to make it clear to emphasize the work done exactly by the authors.
2. The authors used dilation, pyramid pooling and all sorts of upsampling schemes to handle the issue of small object detection. But the above techniques actually did not give any new information regarding the small object but only avoiding losing them. Probably, there are some more straightforward strategies that can deal with this problem rather than to consider or to improve a sub-optimal model.
3. Figure 2 compared the detection accuracy of different sizes of objects. It would be better to define exactly the small object, i.e., what are the exact sizes for small objects, medium objects, and large objects.
4. Table 1 gave the results of detection for different methods. It would be useful to tell readers the detailed numbers for the experiment, such as the number of the training pictures, the training and inference details for each of the benchmarked methods.
Author Response
The co-authors and I would like to thank you for the time and effort spent reviewing the manuscript. We sincerely thank the reviewer for the comments. We have updated our response letter. Please see the attachment.

Reviewer 2 Report
Comments and Suggestions for Authors
The paper introduces ERF-RTMDet, which is a method designed to improve the detection of small objects in remote sensing images. The proposed model uses dilated spatial pyramid pooling, CARAFE upsampling, and hybrid dilated attention to enhance the receptive field and context information of small objects. The paper claims that the method improves the detection accuracy of small objects without compromising the performance on medium and large objects. The method is tested on two datasets: MAR20 and NWPU VHR-10.
Some minor comments to the authors:
1. The ablation study presented in your paper is very interesting as it demonstrates how different improvements affect accuracy compared to the baseline. It provides valuable insights into the effectiveness of each component.
2. If you can get access to a computation cluster, it would be intriguing to explore the impact of using a larger input size, similar to how models like YOLOv8 and Faster-RCNN have shown to yield good results on small objects with input sizes like 1280x1280.
3. You should confirm whether you used the same image input size during inference as during training. Additionally, in chapter 4.2, please correct the "x" symbol for the image size, for clarity.
4. Since you are downsizing the input images anyway, it would be informative to include data on the average size of objects in both datasets as a percentage of the total image size. This context can help readers better understand the scale of objects you are dealing with.
5. Regarding your training process, did you evaluate the models after 200 epochs, or did a "best model" emerge at an earlier stage? Did you perhaps also use some early stopping criteria during training? Please clarify.
6. It's been shown in other research, that increasing the number of classes may positively impact overall model accuracy. Have you considered testing edge cases with a very small number of classes (even just one class) and a large number of classes if an RS dataset with such characteristics is available? This analysis could shed light on the model's scalability and versatility.
7. Consider also testing and comparing on more modern versions of YOLO detectors (e.g. YOLOv7 and YOLOv8).
Author Response

(The authors gave the same response as above.)

Reviewer 3 Report
Comments and Suggestions for Authors
The authors propose a ERF-RTMDet can both obtain better detection results on small objects and maintain for slightly boost the detection effect on mid-scale and large-scale objects. Yet, it is not clear what the authors consider as small objects. The scale of the figures 8, 9 and 10 is essential for validating this issue.
Lines 138-156 This section for the methods must be enhanced with the experience from the use of these methods in the detection process and also one phrase about the technique that these methods are based.
Line 330-333: The authors claim that their method is best for mAP and mAP_s on both NWPU VHR-10 and MAR20 datasets. Similarly for Results on NWPU VHR-10 dataset: Compared with representative one-stage and two-stage object detection methods. Yet the conclusions miss important statement on the expectations of the detection results in future dataset (with moving objects) or in past RS images where the resolutions of the images might not be so high. It is expected by the reader to know the minimum resolution requirements on which this approach will furnish similar results.
It is also essential to explain at the caption of each figure what are the various sections (fig.1, a, b and c), and at the flow chart figures 3, 4, 5, 6 and 7 what are the acronyms that are used in each box. The captions must be self-explanatory and independent of the text.
In general,, after the introduction of these amendments the text should be considered ready for publication. This is valid only after the successful incorporation of the essential text.
Comments on the Quality of English LanguageThe quality of the English is good.
Author Response

(The authors gave the same response as above.)

Reviewer 4 Report
Comments and Suggestions for Authors
The topic of the reviewed article is highly topical. Developing the methods to enhance the effect of detecting small objects in remote sensing (RS) datasets is urgently needed. The paper is suitable for publication in the Journal: Remote Sensing.
The authors of the paper proposed a small object detection method with an enhanced receptive field, ERF-RTMDet. The proposed ERF-RTMDet method is characterized by a more robust ability to detect small objects in RS images. Comprehensive experimental results confirm the effectiveness of ERF-RTMDet on publicly available MAR20 and NWPU VHR-10 datasets. Compared with other most representative models, ERF-RTMDet can obtain better detection results on small objects and maintain or slightly increase the detection effect on medium-sized and large objects.
I have no comments on the reviewed article.
I evaluate the article as an interesting scientific article that brings new scientific knowledge in the field of increasing the efficiency and further specialization of the YOLO algorithm.
Author Response
We sincerely thank the reviewer for the comments. The coauthors and I would like to thank you for your recognition and comments on our manuscript and research. Please see the attachment.

Round 2
Reviewer 1 Report
Comments and Suggestions for Authors
Thanks for the authors, who took my comments seriously and responded carefully. Majority of my concerns have been addressed satisfactorily, but there is still one major worry remained, i.e., the point 2, how successful it is by introducing the ERF-RTMDet in the proposed method. The author explained in their response note that the possible improvement came from “the feature maps extracted by our method contain more detailed information of small objects”. But in many deep learning based detection architectures, the idea of multi-resolution idea has been adopted. For example, in SSD, a quite ‘older’ architecture, has used the lowest level feature map that is with the highest resolution and without any pooling. In this sense, it seems that it may be no bother to consider the complicated ERF-RTMDet because there is a much simpler and straightforward solution already existing long ago. Other similar arguments, such as “our method introduces the attention mechanism to avoid complex background interference” and “our method acquires more contextual information about small objects” are also disputable because these mechanisms are less related with the claimed motivation of ‘small object’ detection but would be better put in the category of feature extraction. So I guess the work may need more elaborations prior to be considered in a publication, although I admitted that the authors actually have made a lot of effort to prepare this manuscript.
Author Response
The co-authors and I would like to thank you for the time and effort spent reviewing the manuscript. We sincerely thank the reviewer for the comments. Please see the attachment.
